# Recent Advances in Machine-Learning-Based Chemoinformatics: A Comprehensive Review

**DOI:** 10.3390/ijms241411488

**Published:** 2023-07-15

**Authors:** Sarfaraz K. Niazi, Zamara Mariam

**Affiliations:** 1College of Pharmacy, University of Illinois, Chicago, IL 61820, USA; 2Zamara Mariam, School of Interdisciplinary Engineering & Sciences (SINES), National University of Sciences & Technology (NUST), Islamabad 24090, Pakistan; zmariam.msbi21rcms@student.nust.edu.pk

**Keywords:** QSAR, QSPR, chemoinformatics, small molecules, AI/ML, molecular descriptors, biological activity, SAR, predictive modeling, computational validation

## Abstract

In modern drug discovery, the combination of chemoinformatics and quantitative structure–activity relationship (QSAR) modeling has emerged as a formidable alliance, enabling researchers to harness the vast potential of machine learning (ML) techniques for predictive molecular design and analysis. This review delves into the fundamental aspects of chemoinformatics, elucidating the intricate nature of chemical data and the crucial role of molecular descriptors in unveiling the underlying molecular properties. Molecular descriptors, including 2D fingerprints and topological indices, in conjunction with the structure–activity relationships (SARs), are pivotal in unlocking the pathway to small-molecule drug discovery. Technical intricacies of developing robust ML-QSAR models, including feature selection, model validation, and performance evaluation, are discussed herewith. Various ML algorithms, such as regression analysis and support vector machines, are showcased in the text for their ability to predict and comprehend the relationships between molecular structures and biological activities. This review serves as a comprehensive guide for researchers, providing an understanding of the synergy between chemoinformatics, QSAR, and ML. Due to embracing these cutting-edge technologies, predictive molecular analysis holds promise for expediting the discovery of novel therapeutic agents in the pharmaceutical sciences.

## 1. Introduction

In 1998, the term “chemoinformatics”, coined by Frank K. Brown, was intended to hasten drug discovery and development; however, now, chemoinformatics is crucial in biology, chemistry, and biochemistry. The general process of drug discovery took 12 to 15 years and involved investments of around $500 million in 1998. New developments in machine learning (ML) and artificial intelligence (AI) have revolutionized chemoinformatics and drug discovery to a great degree. Market revenue for small-molecule drug discovery was $75.96 billion in 2022 and is projected to hit around $163.76 billion by 2032 [1,2].

In contrast to previously well-established statistics, mathematics, and physics-based stand-alone models, ML has introduced a paradigm shift, allowing computers to analyze data and draw conclusions and predictions without relying solely on explicit rules or predefined mathematical equations. These algorithms can discover complex patterns and relations in 3D chemical structures and biological activity data, adaptively adjust their models based on feedback, and generalize from training examples to make accurate predictions on unseen data. This data-driven approach has opened new avenues for optimizing drug–target interactions; empowering target-based drug discovery, chemical library screening, molecular modeling, mechanics, and dynamics; prioritizing potential drug candidates; and predicting possible toxicological responses of biologics with improved accuracy and efficiency. This review discusses the current state of research, the potential integration of ML-driven chemoinformatics tools, techniques in drug discovery, and the challenges and limitations of using these methods. Through a comprehensive analysis of recent studies and developments, we aim to provide insights into the exciting possibilities this integration holds for the future of small-molecule drug discovery and design.

## 2. Exploration of Chemoinformatics

At the intersection of chemistry and informatics, chemoinformatics has emerged as a potent field in drug discovery, employing inductive learning to predict chemical phenomena [3,4]. With the exponentially increasing accessibility of chemical data, the application of ML in chemoinformatics has revolutionized the way researchers now explore, analyze, and predict the properties and activities of molecules. Compared to a few decades ago, it has expedited the process by many folds. It focuses on molecular engineering, molecular manipulation, library design, compound database searching, chemical space exploration, molecular graph mining, pharmacophore, and scaffold analysis [5,6,7,8,9]. 

## 3. Fundamentals of Chemoinformatics

ML models perform prediction tasks based on chemical training data provided in the form of mathematical equations or a numerical representation. This transformation of compound structures into machine-learning-ready chemical data involves a complex, multilayer computational process. The process encompasses descriptor generation, molecular graphs, fingerprint construction, similarity analysis, chemical space searching, molecular dynamic simulations, etc. Each layer is interwoven with the preceding layers, significantly influencing the interpretation of the chemical data by the machine learning models and enhancing their predictive capabilities.

### 3.1. Data Mining and Chemical Databases

Training ML models requires chemical data, and chemoinformatics involves using chemical databases to store and retrieve chemical information. These databases enable searching for specific molecules or analyze large chemical datasets. The training of models relies heavily on managing and utilizing chemical databases that store vast amounts of chemical information, including compound structures, biological activities, and other relevant physiochemical properties. These databases facilitate data mining, knowledge discovery, and information retrieval for target prediction. Specialized databases of naturally existing compounds, including LOTUS [10], COCONUT [11], SuperNatural-II [12], NPASS [13], SymMap [14], TCMSP [15] and TCMID [16] provide valuable resources. These databases contain comprehensive information on compound structures, molecular physicochemical properties, and molecular descriptors.

Utilizing the known structures of these compounds, abductive techniques based on structural similarities can be leveraged to convey knowledge regarding the mechanism. Various similarity scores, as mentioned before, can be computed, considering the similarity of 1D structures (e.g., SMILES- or SELFIES-based similarity [17]), 2D structures (e.g., 2D fingerprints or topological similarity), and even 3D structures (e.g., 3D geometric shape-based similarity). Previous studies have identified several metrics suitable for molecular similarity calculations, including the Tanimoto index, Manhattan distance, Dice index, overlap coefficient, cosine coefficient, and Soergel distance [18,19,20]. Furthermore, chemical bioactivity and structural data can be acquired from drug databases like ChEMBL [21], BindingDB [22], DrugBank [23], Inxight [24], and Protein Data Bank [25]. Despite the availability of extensive databases, utilizing machine learning and deep learning techniques offers significant potential to enhance the creation of molecules and focused libraries, enabling the discovery of potent bioactive compounds through targeted design and generation strategies in QSAR studies.

Generative models like recurrent neural networks (RNN) have been employed to generate novel chemical structures predicted to have desirable properties, such as high potency or low toxicity. RNN models have been previously used to generate focused molecule libraries and have implicitly learned chemical knowledge to create molecules with combined characteristics of both bioactive natural products and synthetic compounds, such as DeepMGM. Besides this, generative models have been used for inverse QSAR/QSPR, which involves generating molecules that meet specific target properties.

The DeepMGM model was trained using drug-like molecules and produced a general model (g-DeepMGM) capable of generating scaffold-focused libraries. A target-specific model (t-DeepMGM) for the cannabinoid receptor 2 (CB2) using transfer learning was also developed. A discriminator was incorporated into DeepMGM for in silico molecular design and testing. The generated molecule XIE9137 was identified as a potential CB2 allosteric modulator, highlighting the effectiveness of deep learning in de novo molecular design and chemical library generation [26,27].

### 3.2. Chemical Data Representation

Advancements in ML modeling and the availability of a vast pool of chemical and biological data have led to a dire need for data to be translated into computer-understandable form before models are trained on them. Chemical data representation can be empirical, molecular, and structural data represented in molecular graphs, fingerprints, descriptors, etc. [28,29]. A multivariate random forest model generated for genomic characterization was trained on genomic sequencing data given in numerical representation in one study [30]. In another, a Naïve Bayesian (NB) model was developed on numeric-based activity data, representing antagonists’ binding on estrogen receptors [31]. An ML-based model was trained on 31 chemical numerical datasets obtained from Merck to predict the properties of small compounds based on ADMET (absorption, distribution, metabolism, excretion, and toxicity) [32]. Similarly, molecular fingerprint data have also been used to train such models for ADMET properties prediction. NB and QSAR integrated models have been used to predict active compounds against human immunodeficiency virus type-1 trained on descriptors including extended-connectivity fingerprint data [33]. Furthermore, the graph neural networks (GNNs) function with the graph structure data of 3D molecules and have been used to identify potential drug molecules [34]. Besides the choice of representation, data augmentation, and pre-processing, the twin curse of dimensionality and collinearity must be tackled.

When encountered in these data representations and modeling approaches, the twin curse of dimensionality and collinearity is addressed through principal components analysis (PCA), partial least squares (PLS), and other available techniques. The data often involve many genomic or chemical descriptors in genomic characterization and small-molecule property prediction. This high-dimensional feature space can lead to overfitting, decreased model interpretability, and increased computational complexity. In studies involving activity data, binding assays, or molecular fingerprints, collinearity can arise from strong correlations or dependencies among these input variables. Highly correlated variables can introduce redundancy and multicollinearity issues, leading to unstable model estimates and difficulties in interpreting the contributions of individual variables.

To address these challenges, dimensionality reduction techniques such as feature selection, feature extraction, data regularization, penalization, and genetic algorithms can help mitigate these issues by imposing constraints and encouraging sparsity. The principal components analysis (PCA) and the partial least squares (PLS) methods generally transform massive datasets with correlated variables into smaller uncorrelated ones. PCA has been used to explore complex datasets in QSAR and dimensionality reduction. A study investigating PCA’s different applications in QSAR uses a dataset including CCR5 inhibitors. PCA has been used to detect outliers in the datasets, as well. The original data matrix from a different investigation was examined using PCA, in which molecules are represented by several predictor variables (molecular descriptors). PCA has also been used to design features for estrogen receptor binding prediction. Furthermore, observations revealed enhanced performance in therapeutic activity predictions against a diverse range of pharmacological protein targets identified by the kernel–principal components (kernel-PCA) analysis and a nonlinear PCA variation, surpassing the predictive capabilities of LASSO regression.

Similarly, the partial least squares (PLS) method has been employed to discern significant structural patterns that contribute to the biological activity of a molecule. The efficiency and accuracy of PLS in combination with unsupervised dimensionality reduction techniques surpass the approach of explicitly combining unsupervised dimensionality with multivariate regression. PLS is also widely utilized in the field of 3D-QSAR modeling [6,35,36].

### 3.3. Molecular Descriptors

Molecular descriptors are quantifiable representations that capture chemical compounds’ structural, physicochemical, and biological properties. These descriptors are quantitative measures used for similarity analysis, virtual screening, and predictive modeling. Chemical molecular descriptors are categorized as 0D, 1D, 2D, 3D, and 4D (Table 1) [37,38,39,40].

0D Descriptors: These are constitutional or count descriptors, scalar values that describe several atoms, bonds, or functional groups in the molecule, e.g., molecular weight.1D Descriptors: These descriptors capture molecular properties in one dimension along a linear sequence or chain of atoms, e.g., structural fragments or fingerprints.2D Descriptors: These descriptors provide information about the structure on a molecular level and its properties within a 2D plane, e.g., topological polar surface area (TPSA) and graph invariants.3D Descriptors: These descriptors describe the molecular properties in 3D space, considering the spatial arrangement of atoms, e.g., autocorrelation descriptors, substituent constants, surface:volume descriptors, quantum, chemical descriptors, 3D-MoRSE descriptors, WHIM descriptors, GETAWAY descriptors, size, steric, surface, and volume descriptors.4D Descriptors: These descriptors encompass properties that change over time or involve spatiotemporal aspects, e.g., drug dissolution rate, Volsurf, and GRID or CoMFA methods.

These molecular descriptors have been used to select the most relevant properties. MoDeSus is an ML-based tool used to determine the most informative molecular descriptors for QSAR studies. Molecular descriptors allow for ligand-based scaffold hopping for hit and lead optimization, which speeds up the early stages of drug development and has been used to compare QSAR and QSPR models. Although each type of descriptor plays a vital role, 3D and 4D descriptors have shown the most significant contribution to identifying active molecules and potential drug targets. Furthermore, 4D descriptors like CoMFA and GRID have been used to identify active sites of receptors and characterize interactions providing insight into the functional properties of small molecules [41,42,43].

## 4. QSAR

Based on its physicochemical characteristics, a ligand’s biological response or activity can be predicted using QSAR analysis [38]. QSAR modeling techniques have been used to find prospective drug candidates, and these have developed into AI-based QSAR methods [44]. Modern machine learning approaches can be applied to model QSAR or quantitative structure–property relationships (QSPR) and create predicative models based on artificial intelligence [45,46]. Chemoinformatics, QSAR, and machine learning applications have been used to showcase different structure-based, ligand-based, and machine-learning-based approaches for drug development. QSAR/QSPR models employ information on multiple levels, e.g., chemical data, descriptors, molecular graphs, fingerprints, similarity analyses, and molecular dynamic simulations, to predict the most optimal properties of a potential drug.

Structure–activity relationship (SAR) analysis investigates how the chemical structure of a compound relates to its biological activity or properties and plays a crucial role in exploring potential effects of bioactivity on changes in the chemical structure of drugs. Quantifying the degree of the structural or chemical similarity between molecules and extrapolating chemical attributes from molecular similarity are the goals of similarity analysis [47]. Similarity search mainly aims to identify compounds with similar bioactivity to a reference molecule but with different chemotypes. This results in scaffold-hopping derivatives acquired from a reference compound with a novel core structure. Fragment replacement approaches, fingerprint-based similarity search, pharmacophore matching, and 3D shape-based similarity search are all examples of computational research for scaffold hops. Designing molecules of novel scaffolds with increased pharmacological activity and identical 3D structure but a multimodal deep transformer neural technique for scaffold hopping aids the distinct 2D structure [48,49,50,51].

SARs are also employed in clustering, inter-molecular comparisons, outlier and novelty analysis, diversity quantification, and outlier analysis. Molecular datasets are used by AIMSim, a unified platform, to carry out similarity-based tasks using binary similarity metrics and molecular fingerprints [52]. In a study, a tool called the similarity ensemble approach (SEA) was used to estimate the accuracy of k-nearest neighbors (kNN) QSAR models constructed for known ligands of each GPCR target individually to discover active and inactive molecules [53]. ChemSAR, another tool, offers an integrated web-based platform for creating SAR classification models, and it is also an online pipelining platform for molecular SAR modeling. For the identification and structural organization of analog series, SAR analysis, and compound design, various researchers have employed the SAR Matrix (SARM) concept [54,55,56]. DeepSARM, which combined deep learning and generative modeling, was introduced, expanding the scope of the SARM technique. This improvement made it possible to create target-based analogs by considering the chemical information from related targets to increase structural uniqueness and variety [57]. 

The current approach to constructing QSAR models typically involves generating descriptors for the compounds in the training set, applying descriptor selection algorithms, and employing statistical fitting methods to build the model. Nevertheless, there have been investigations into the potential for developing high-quality, interpretable QSAR models for large and diverse datasets without relying on pre-calculated descriptors. To achieve this objective, these studies explore using deep learning techniques, specifically long short-term memory neural networks [58].

### 4.1. QSAR Modeling

The standardized procedure for building QSAR models in drug discovery encompasses a series of modular steps that incorporate the afore-discussed chemoinformatics and machine learning techniques. By following the protocol, QSAR modeling aided by ML and DL (deep learning) can predict the properties or activities of chemical compounds, toxicity, and other related physiochemical properties.

### 4.2. Molecular Encoding

Molecular encoding is like chemical data representation transformation, as discussed before. Compounds’ chemical characteristics and attributes are directly deduced from their chemical structures or by looking up experimental findings. This process involves extracting relevant information from the molecular structure, such as atom types, bond types, functional groups, and physicochemical properties.

### 4.3. Feature Selection

Feature selection in QSAR aims to identify the most informative and relevant features from a larger set. It involves techniques such as univariate analysis, filter methods, wrapper methods, and embedded methods. To determine the most pertinent attributes and lessen the dimensionality and collinearity of the feature vector, hybrid feature selection, feature learning methodologies, and unsupervised learning techniques are applied. These techniques have successfully preserved a fair computing effort without reducing the precision of the final QSAR models [59].

### 4.4. Model Training

During the model training and learning phase of QSAR modeling, a supervised machine learning model is generally employed to uncover an empirical function that effectively maps input feature vectors to biological responses. This function is optimized to achieve the best possible mapping. It is crucial to carefully select and consider the SAR datasets and descriptors used for training and model validation to ensure the development of accurate QSAR models [60].

## 5. Machine-Learning-Based QSAR Modeling

Unsupervised learning and supervised learning are two categories of machine learning models. In supervised learning, a model is trained with labeled data to produce predictions based on known input–output correlations (for example, support vector machines and linear regression). Unsupervised learning analyzes unlabeled data to discover underlying patterns and relationships without explicit guidance (e.g., clustering and dimensionality reduction). 

QSAR involves training supervised learning models using labeled datasets, where the input features represent the chemical structures and the output labels represent the corresponding biological activities, toxicity, or other properties. Furthermore, unsupervised learning techniques can be applied in QSAR to uncover hidden patterns or relationships within the chemical data, such as clustering similar compounds based on their structural similarities or reducing the dimensionality of the dataset. QSAR models can be built using traditional methods like random forest, multiple linear regression, Naïve Bayes, k-nearest neighbors, support vector machine, or deep neural network (DNN). 

The hit-to-lead optimization and hit-to-hit identification can be performed using ML-based QSAR models. The automated hit identification and optimization tool (A-HIOT), among other sophisticated virtual screening frameworks, can be employed to find and improve hits for fixed protein receptors. A-HIOT uses numerous open-source methods to combine chemical and protein space and produce high-quality predictions [61]. To show that deep neural networks (DNN) and random forests (RF) were superior in hit prediction efficiency, comparison studies between DNN and other ligand-based virtual screening (LBVS) approaches were conducted. A scan of an in-house library of 165,000 chemicals revealed numerous triple-negative breast cancer (TNBC) inhibitors as powerful hits using DNN. From a small-size training set of 63 compounds, a potent mu-opioid receptor (MOR) GPCR proteins agonist was found. One hundred seventy-six possible antimalarial hits were found by integrating QSAR and virtual screening [62,63].

### 5.1. Regression Analysis

Regression analysis is a statistical technique for simulating the relationship between a dependent variable and one or more independent variables. It seeks to identify the line with the best fit that minimizes the sum of the squared residuals. The relationship between variables can be inferred by estimating the regression equation coefficients. Early QSAR techniques like Hansch and Free Wilson analysis heavily utilized multivariate linear regression. Since QSAR deals with multidimensional data, the twin curses must be tackled before further processing chemical data. Many variations and ensembles of regression analysis are now employed for predictive modeling in QSAR.

By fusing aspects of network analysis and piecewise linear regression, interpretable QSAR models have been created using network-based linear regression. In a study on inhibitors of polo-like kinase-1 and linear regression, to find prediction models of an extensive and structurally varied dataset of 530 chemicals, QSAR models were created. A discriminant–regression model (DIREM), a common discrete–continuous QSAR technique, is another form of regression model that combines discriminant and regression studies to investigate structure–activity connections for substances. The effectiveness of PLS-based QSAR models was assessed, and they were compared with the outcomes of multiple linear regression (MLR) and principal component regression (PCR) in a comparative analysis study on 5-nitrofuran-2-yl derivatives as inhibitors of Mycobacterium TB H37Rv. Compared to PCR, the findings of the PLS and MLR analyses demonstrated significantly higher predictive power and reliability, attesting to the dependability of these techniques [64,65,66,67,68]. Although numerous medication optimization studies have successfully used linear regression analysis and its derivates, there are still substantial drawbacks, including underlying linearity, overfitting, restricted interpretability, the necessity of high-quality data, and false vector space assumptions.

### 5.2. K-Nearest Neighbor

The k-nearest neighbors (kNN) algorithm represents labeled and unlabeled data nodes in a multidimensional feature space. The k-nearest neighbors (kNN) methodology is a straightforward distance-learning strategy in which an unknown member is categorized based on most of its k-nearest neighbors. Using a majority-voting rule, it assigns labels to query points by transferring them from the nearest neighbors. This approach leverages the proximity of data points in the feature space to make predictions [69].

Choosing the right number of nearest neighbors to utilize in the kNN algorithm can be challenging because doing so can lead to unfavorable false-positive or false-negative rates. This was addressed by introducing the similarity ensemble approach (SEA), which proved to be a more organized method for determining the right number of neighbors in kNN analysis. The SEA compares chemical similarity values to a randomized background score, similar to the BLAST sequence similarity search method [70].

A study developed a 3D QSAR model for 30 drugs with anti-HIV activity using a kNN model. According to most of the training set’s nearest neighbors, this kNN model categorized the chemicals. The technique determined the essential structural characteristics that lead to the compounds’ anti-HIV efficacy [69]. Another study developed QSAR models for 50 compounds with an anti-HIV activity using the kNN–molecular field analysis method. The results showed the importance of electrostatic and steric interactions in influencing the anti-HIV activity of the compounds [71]. Consensus kNN QSAR was used in a study and proved to be a practical method for quickly screening the estrogenic activity of organic compounds. It is a flexible method for predicting the estrogenic activity of organic compounds in silico [72]. Utilizing a deep neural network in conjunction with the kNN approach to creating QSAR models for a collection of 1000 chemicals having anti-cancer activity was also claimed to be helpful. According to the study, the main structural characteristics contributing to the compounds’ anti-cancer efficacy could be determined using the kNN approach [73].

### 5.3. Naïve Bayes

Naïve Bayes is a probabilistic classifier commonly assuming that features are independent, simplifying the modeling process. It determines the probability of correct label assignment by considering the prior probability distribution of labels in the training set. It assumes conditional independence between multiple labels and calculates probabilities for each label individually. The PASS program, a notable example, utilizes this approach for predicting drug activities [74].

On 18 sizable, varied in-house QSAR datasets, a study examined the capacity of Pipeline Pilot Naïve Bayes (PLPNB) and random forest to produce precise predictions. According to the study, PLPNB could predict binary and multicategory activities with accuracy and was computationally efficient. Large-scale virtual screening for important pharmacological features, such as cytochrome P450 inhibition, human plasma protein binding, and animal model bioavailability, have demonstrated their effectiveness [75,76]. Another study showed that when used in QSAR modeling, the Naïve Bayes model delivers the lowest mean error when the data points are distributed uniformly. This study uses QSAR as an example to demonstrate the Naïve Bayes model’s optimality [77]. In a comparative study to choose the best learning algorithm and optimal feature selection, Naïve Bayes was shown to be one of the best-performing algorithms for small datasets [78].

### 5.4. Support Vector Machine

Support vector machines (SVM) are widely used in QSAR due to their ability to handle high-dimensional data and nonlinear relationships. They construct a hyperplane that maximally separates different classes in the feature space. SVMs have demonstrated excellent performance in various QSAR applications, such as predicting compound activities, toxicity, and bioavailability. Their versatility and robustness make them valuable tools in QSAR modeling. A framework known as “ML-QSAR” was established in a study in which machine learning methods were used for QSAR modeling. SVM was discovered to be one of the most often used machine learning algorithms in QSAR modeling. The framework was created to facilitate the selection of appropriate strategies among existing algorithms according to the application area requirements and to help develop and improve current approaches [79].

A study developed multiple QSAR methods using several ML algorithms, including SVM, to predict the activity of active substances against Pseudomonas aeruginosa. The study found that SVM could better predict the compounds’ activity accurately compared to other models [80]. Another study examined SVM’s effectiveness and prognostication power in HEPT derivative QSAR modeling. This investigation showed that SVM outperformed different approaches, including artificial neural networks, in terms of prediction [81]. SVM was also used to simulate phenethylamines’ structure–activity relationships (SAR). To categorize antagonists and agonists and forecast their effects, the study used SVM, which it discovered to be a reliable method in the SAR/QSAR field [82]. 

Another study evaluated the effectiveness of 16 machine learning algorithms, including SVM, on 14 QSAR datasets and concluded that various ML algorithms offered different QSAR modeling approaches to uncover the connections between compound structures and properties [83]. When used for large-scale ligand-based predictive modeling, SVM predicts the properties of new, unknown compounds and can achieve good predictive performance for large-scale QSAR modeling [84]. SVMs have also been applied in a QSAR investigation involving ethyl 2-[(3-methyl-2,5-dioxo(3-pyrrolidinyl)) amino]-4-(trifluoromethyl) pyrimidine-5-carboxylate derivatives, targeting the transcription factors activator protein (AP)-1 and nuclear factor (NF)-kB [85]. To determine the structural elements that give aminopyrimidine-5-carbaldehyde oxime derivatives a potent vascular endothelial growth factor (VEGF)-2 inhibitory action, a genetic variable selection approach was combined with SVMs. This integrated approach successfully identified several critical structural features associated with the desired biological activity, proving SVM helpful in QSAR modeling [86].

### 5.5. Convolutional Neural Networks, Recurrent Neural Networks, Deep Neural Networks, and Ensemble Methods

By leveraging the power of neural networks with multiple hidden layers, deep learning models can effectively learn complex relationships between molecular structures and their related biological activities. In QSAR, deep learning models, such as convolutional neural networks (CNNs), recurrent neural networks (RNNs), and deep neural networks (DNNs), have been utilized to analyze and predict various properties of molecules, including binding affinity, activity, toxicity, and bioavailability. These models and their ensemble methods have been applied in QSAR studies to enhance models’ accuracy and predictive power. 

CNNs have successfully captured molecular features and patterns from 2D chemical structures and search spaces. RNNs have been utilized to model sequential data, such as molecular fingerprints and SMILES strings. DNNs have effectively learned complex relationships between 3D and 4D molecular descriptors and their respective bioactivity data. Ensemble methods combining CNN, RNN, and DNN have been employed to improve prediction performance. These advanced neural network topologies and ensemble methods have been extensively used in modeling QSAR/QSPR features of small compounds and conducting pharmacokinetic and pharmacodynamic studies, alongside work in other fields of chemoinformatics. In particular, CNN’s unmatched capacity for image analysis made it possible to visualize protein structures as ‘3D images’ with four separate atom-type channels. These 3D-CNNs were used to compare the microenvironments of amino acids and predict how mutations might affect the structure of proteins [87]. A Transformer–CNN architecture was suggested in a study for QSAR modeling and interpretation. Convolutional and element-wise feed-forward layers were used in place of all recurrent units in the design, and it was discovered that the Transformer-CNN architecture produced good results for small datasets and converged quickly for QSAR tasks [88].

Recurrent neural networks (RNNs), also known as long short-term memory (LSTM) networks, are built to recognize both short-term and long-term dependencies in sequential input. For applications like de novo drug design, where they learn the structural patterns and rules from SMILES strings to produce novel molecules, LSTM networks have been used in the context of QSAR. Deep reinforcement learning, variational autoencoders, and generative adversarial networks (GANs) are other cutting-edge methods used to generate compounds with precise molecular features while learning latent representations of molecules. These methods aid in the discovery of novel medication candidates and the exploration of new chemical territory [27,89,90,91,92,93]. A study that proposed an ensembled RNN–CNN architecture, DeepCpG, for DNA methylation analysis concluded that combining RNN and CNN improved the performance of the QSAR model [94]. To perform QSAR analysis utilizing three-dimensional photographs of chemical structures, a brand-new DL-based method dubbed DeepSnap was created. Without extracting descriptors, this method may also forecast the potential toxicity of many compounds to different receptors. To perform QSAR analysis utilizing three-dimensional photographs of chemical structures, a brand-new DL-based method dubbed DeepSnap was created. Without extracting descriptors, this method may also forecast the potential toxicity of many compounds to different receptors [95]. CNN, RNN, and deep-learning-based methods have also shown promising results in QSAR modeling.

## 6. Validation of ML-QSAR Models

ML-QSAR models are typically assessed using established metrics like sensitivity, specificity, precision, and recall. In cases where the dataset is unbalanced, the area under the curve (AUC) obtained from receiver operating characteristic (ROC) curves can be employed. QSAR models can also be evaluated by various methods, such as external validation, conformal prediction methods, and evaluation of QSAR equations for virtual screening. External validation is the primary method for evaluating the accuracy of generated models for the activity prediction of compounds that have not yet been synthesized. Understanding the variables that control molecular characteristics and creating new compounds with advantageous features depend on QSAR models, which provide information on the association between activities and structure-based molecular descriptors [96,97].

Even though 3D-QSAR techniques like CoMFA take structural conformation into account, they are computationally intensive and can introduce errors related to conformation prediction, ligand orientation, and structural alignment. Consequently, 2D-QSAR models can provide a viable alternative and sometimes even outperform 3D-QSAR strategies [38]. The creation of verified models for accurate and precise prediction of a compound’s biological actions is the ultimate goal of QSAR analysis. When creating QSAR models, metrics like R2 and QCV2 are generally optimized. The performances of the final models are assessed using comparable metrics computed on external datasets [98].

A comparative study on 5-nitrofuran-2-yl derivatives as inhibitors of Mycobacterium tuberculosis H37Rv used statistical parameters, including squared correlation coefficients, cross-validated correlation coefficients, and Fischer’s value for statistical importance, to assess the quality of the generated QSAR models. Another study examined several statistical parameters of 44 published QSAR models for biologically active substances that were externally validated and presented in academic journals. They concluded that using the coefficient of determination (R2) alone was insufficient to determine if a QSAR model was viable. There are benefits and drawbacks to these defined criteria for external validation that should be considered in QSAR investigations [99].

## 7. Interpretability and Explainability of ML-QSAR Models

The creation of verified models for accurate and precise prediction of a compound’s biological actions is the ultimate goal of QSAR analysis. The interpretability and explainability of ML-QSAR models promote transparency, reproducibility, and trust in the models’ predictions, allowing researchers and stakeholders to make informed decisions regarding drug discovery and development. Six artificial datasets of varying degrees of complexity were produced as part of a study to compare various QSAR model interpretation techniques. These datasets were used in the study’s investigation of a wide range of descriptor and algorithm pairings and the Structure–Property Correlation Index (SPCI) method of universal interpretation. The study showed that predictivity might decline more quickly than interpretation performance and that even models with good predictivity may occasionally have subpar interpretation performance [100].

Various techniques can enhance the explainability and interpretability of ML-QSAR models. Feature importance analysis can identify the most influential molecular descriptors or features contributing to the model’s predictions. Visualization methods, such as heat maps or feature importance plots, can aid in understanding the relationships between features and the predicted outcomes. Additionally, model-agnostic techniques like LIME (Local Interpretable Model-Agnostic Explanations) [101] or SHAP (Shapley Additive Explanations) [102] can provide insights into individual predictions by highlighting the contributions of each feature. A new way to visualize QSAR models is described in a publication that streamlines analysis by adding a new measure of model similarity. The method relies on projecting models into a two-dimensional plane, where the separation between two models is proportional to the variation in their expected activities [103]. Another study creates predicted QSAR models that may be projected onto the atoms of a molecule by combining direct kernel-based PLS with Canvas 2D fingerprints. The work offers a model visualization that can be used to determine which atoms are most important for forecasting activity [104].

Being unable to explain why a neural network generates a prediction is a significant impediment to the application of AI models due to the ‘black box’ approach. In addition to discouraging chemists from utilizing deep learning predictions, this has caused neural networks to pick up undetectable bogus correlations. Counterfactual interpretation is a technique for reading ML models that can be used to comprehend why a model generates a specific prediction. Counterfactuals are local interpretations that can disclose the contributions of atoms or fragments within particular molecules to identify the most beneficial or detrimental motifs to consider for future alterations in the context of QSAR models. Because they resemble counterfactuals, instance-based techniques have been claimed to provide ‘natural’ model interpretations for researchers. Various approaches to interpretation have been established; however, there are no appropriate standards to assess how well they apply to the interpretation of QSAR models. An approach known as STONED (Structure–Topology Optimization for Novel Explanatory Discoveries) is suggested in a study; it produces molecular counterfactuals for any model. These molecular counterfactuals offer skeletal, molecular structure-based explanations. All molecules produced by STONED are legitimate substances, so the method does not require training a counterfactual generator. This simplifies the procedure and eliminates the necessity of a generative counterfactual creator [100,105,106].

## 8. Conclusions

Applying machine learning techniques in chemoinformatics has contributed significantly to discovering and designing highly effective drugs. This paper highlights the significant role of chemoinformatics and ML-based QSAR in drug discovery and development. Integrating computational approaches with large-scale data analysis has revolutionized the field, enabling efficient exploration of chemical space and predicting biological activities. Multiple algorithms built for QSAR modeling significantly highlight features necessary for further designing small molecules. They have demonstrated their effectiveness in predicting molecular properties and activities, aiding in compound prioritization and optimization.

The future of chemoinformatics and QSAR modeling holds promising opportunities for further advancements. Integrating QSAR models with molecular docking techniques can enhance the accuracy of binding affinity predictions and provide valuable insights into the interaction between ligands and target proteins. Fragment-based design approaches can benefit from QSAR models by guiding the selection and optimization of fragments to develop novel drug candidates. Additionally, integrating QSAR models with de novo drug generation methods, such as deep learning and generative modeling, opens up possibilities for computer-assisted design and discovering new molecules with desired properties.

This convergence of QSAR models with molecular docking, fragment-based design, and de novo drug generation methods holds great potential to accelerate the drug discovery process, reduce costs, and increase the success rates of identifying novel therapeutic agents. Continued research and development in this area will undoubtedly pave the way for more efficient and precise drug design strategies, ultimately benefiting patients and advancing the field of pharmaceutical sciences.

## Figures and Tables

**Table 1 ijms-24-11488-t001:** The most common 0D to 4D chemical descriptors for QSAR/QSPR analysis.

Descriptor Dimension	Descriptor Type	Example
0D	The molecule’s atoms, bonds, and functional groups count	Molecular weight, LogP (partition coefficient)
1D	Molecular properties in a linear manner	Molecular Formula, SMILES & SELFIES
2D	Topological polar surface area (TPSA)	Molecular fingerprint (e.g., Morgan fingerprint), Constitutional descriptors (e.g., atoms, bonds, and rings count)
3D	Special properties of a molecule	Molecular shape descriptors (e.g., volume, surface area), Pharmacophore features
4D	Electrostatic potential descriptors with spatiotemporal aspects	Molecular dynamics descriptors, solvent accessible surface area (SASA), radius of gyration (Rg), Time-dependent properties (e.g., dynamic polar surface area (dPSA), time-dependent dipole moment

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
