# Peer review of "Recent Advances in Machine-Learning-Based Chemoinformatics: A Comprehensive Review"

_ijms, 2023, doi:10.3390/ijms241411488_

Round 1
Reviewer 1 Report
In this paper, the authors provide a review of algorithm developments in cheminformatics starting from rational design and/or physics based strategies to modern machine learning techniques. The review is well written and covers some of the very important developments in the field, but is sparse on generative modeling based approaches of the last 2-3 years.
The authors talk about SMILES as a form of chemical data representation, but they should also discuss the SELFIES representation which ensures structurally valid molecules by construction (details here: https://arxiv.org/abs/1905.13741)
Certain sections can be made more compact and the overall presentation can thus be made more succinct. For instance:
-
In the beginning of section 3.2, the authors mention genomic data just to talk about an application of a random forest model. This appears un-necessary and takes the focus away from the main message of the paper, i.e. developments in cheminformatics.
-
There is not much point in making separate subsections (section 3.4 and section 4) on SAR and QSAR, and much of what is written about featurization and regression modeling in the SAR section reappears later in the QSAR section. Please combine these into a single compact section.
This paper misses important generative modeling efforts such as DiffDock which uses diffusion based methods to hallucinate small molecule structures given the structure of target binding pockets. (details here: https://arxiv.org/abs/2210.017760) It also misses modern explainability efforts for de-novo small molecule design such as the one found here: (https://chemrxiv.org/engage/chemrxiv/article-details/613268f0d5f0803706ba0c79)
I suggest the authors include some of these more modern approaches based on generative modeling to better sync the material to latest developments in the field.
Author Response
We sincerely appreciate your thorough review of our article and your insightful comments. We have carefully considered your feedback and have made the necessary revisions to address your concerns and improve the overall quality of the paper. Thank you for taking the time to help us enhance our work.
Firstly, we have incorporated the SELFIES representation in our discussion of chemical data representation. We now highlight its advantages in ensuring structurally valid molecules by construction. For readers interested in more details, we have included a reference to the paper you mentioned (https://arxiv.org/abs/1905.13741).
To streamline the paper and maintain a stronger focus on developments in cheminformatics, we have removed the mention of genomic data at the beginning of section 3.2. We understand that this information was not directly relevant to the paper's main message and appreciate your suggestion to remove it.
Furthermore, we have combined the separate subsections of SAR and QSAR into a more concise section. By doing so, we have eliminated the repetition of content on featurization and regression modeling, resulting in a more streamlined and coherent presentation.
Thank you for drawing our attention to the important generative modeling efforts using diffusion-based methods for small molecule structure generation. We have included a brief overview of this research direction and referenced the paper you mentioned.
These additions significantly enhance the relevance and completeness of our review article, aligning it with the latest developments in cheminformatics.
We greatly appreciate your time and effort in reviewing our manuscript.
Reviewer 2 Report
Overall, it is a comprehensive review article that summarizes the mainstream applications of cheminformatics in combination of machine learning. The reviewer would suggest an acceptance if the following two comments can be addressed appropriately.
1. In the section of Data Mining and Chemical Databases, it worth describing the application of generative chemistry in creating scaffold-focused and target-specific molecular libraries. It is a fast-growing area of research which deserves a detailed discussion. Related references include but not limited to:
https://doi.org/10.3390/cells11050915
https://doi.org/10.1021/acscentsci.7b00512
https://doi.org/10.1038/s42256-020-0160-y
2. In the section of Machine Learning-based QSAR Modeling, it will be beneficial to briefly touch base on (1) specific applications of using ML for hit finding, and (2) potential platforms established. These kind of intuitions presentations will help readers quickly identify use cases. Examples include:
https://doi.org/10.1038/s41596-021-00659-2
https://doi.org/10.1073/pnas.2000585117
It has been indicated in questions above.
Author Response
We sincerely appreciate your thorough review of our article and positive feedback regarding its comprehensiveness. We have carefully considered your comments and have made the necessary revisions to address your suggestions. Thank you for your valuable input, which has helped us improve the quality and relevance of our review article.
Regarding your first comment, we agree that applying generative chemistry in creating scaffold-focused and target-specific molecular libraries is a rapidly growing area of research. We have now included a detailed discussion highlighting the importance and significance of this field. We explore the advancements and methodologies employed in generative chemistry, emphasizing its contributions to creating diverse molecular libraries.
Regarding your second comment, we understand the importance of providing specific machine learning (ML) applications for hit finding and discussing potential platforms established in QSAR modeling. We have added a brief but informative discussion within the Machine Learning-based QSAR Modeling section to address this. By incorporating this information, we aim to facilitate a better understanding of the diverse applications of ML in QSAR modeling.
Round 2
Reviewer 1 Report
The authors have appropriately addressed all my comments and I recommend publication without any further changes.
Reviewer 2 Report
The reviewer's comments have been addressed appropriately.
N/A